# Prevalence Rates of Depression and Anxiety among Young Rural and Urban Australians: A Systematic Review and Meta-Analysis

**DOI:** 10.3390/ijerph20010800

**Published:** 2023-01-01

**Authors:** Sushmitha Kasturi, Victor M. Oguoma, Janie Busby Grant, Theo Niyonsenga, Itismita Mohanty

**Affiliations:** 1Health Research Institute, University of Canberra, Canberra, ACT 2617, Australia; 2Poche Centre for Indigenous Health, University of Queensland, Brisbane, QLD 4066, Australia; 3Centre for Applied Psychology, University of Canberra, Canberra, ACT 2617, Australia

**Keywords:** rural health, mental health, prevalence, depression, anxiety, adolescence, young people, Australia

## Abstract

Globally, depression and anxiety are major public health concerns with onset during adolescence. While rural Australia experiences overall lower health outcomes, variation in mental health prevalence rates between rural and urban Australia is unclear. The aim of this paper was to estimate the pooled prevalence rates for depression and anxiety among young Australians aged between 10 and 24 years. Selected studies from a systematic literature search were assessed for risk of bias. Random effects model using DerSimonian and Laird method with Freeman–Tukey Double Arcsine Transformation was fitted. Sensitivity analyses were performed. Prevalence estimates were stratified by region and disorder. The overall pooled prevalence of depression and anxiety was 25.3% (95% CI, 19.9–31.0%). In subgroup analysis, anxiety prevalence was 29.9% (95% CI, 21.6–39.0%); depression: 21.3% (95% CI, 14.9–28.5%); and depression or anxiety: 27.2% (95% CI, 20.3–34.6%). Depression and anxiety prevalence were higher in urban 26.1% (95% CI, 17.3–35.9%) compared to rural areas 24.9% (95% CI, 17.5–33%), although the difference was not statistically significant. The heterogeneity was high with an *I*^2^ score of 95.8%. There is need for further research on healthcare access, mental health literacy and help-seeking attitude in Australia.

## 1. Introduction

Mental health is a public health concern worldwide affecting all age groups [1]. Globally, one in six people are aged between 10 and 19 years and one in seven people in this age group experience a mental disorder [1]. Among the mental disorders, depression and anxiety are the most common disorders [1] and are in the 25 leading causes of illnesses and disability worldwide [2]. As of 2021, there are 3.4% and 3.8% of the global population with depression and anxiety disorders, respectively [3]. Globally, adolescent mental health accounts for 13% of the disease burden [1].

In Australia, mental illnesses constitute 13% of the total burden, making it one of the top five leading disease groups in 2018 [4]. One in five Australians aged between 16 and 85 experience some sort of mental disorders in any given year [4]. Evidence shows that the total disease burden on a population is a combination of poor health (non-fatal burden) and loss of healthy life years due to disease and eventual premature death (fatal burden) [4]. Across all age groups, mental disorders contribute to 13% of total disease burden in Australia, and 23% of non-fatal burden making it the second leading cause of non-fatal disease burden [5]. Of this burden, 98% was due to living with the effects of the disorders [5]. In terms of Disability Adjusted Life Years (DALYs), the burden of mental disorders caused 572,775 DALYs in 2015 [5]. 

Australia has 28% (7 million) of its population living in rural and remote areas encompassing various regions and diverse communities [6]. Rural and remote Australia experiences poorer health outcomes largely attributed to social determinants (education, income, employment, and quality of housing) and health risk factors (blood pressure, smoking, healthcare access) [6,7,8,9]. The burden of disease rate in rural areas is 1.4 times higher than that in urban areas. Rural areas reported higher mortality rates (1.4 times), hospitalisations (almost 2 times), prevalence of chronic diseases (cancer, asthma, coronary heart conditions, respiratory conditions) [9] and cost of healthcare access than urban areas, and life expectancy, primary health access such as General Practitioners visits and healthcare workforce are lesser in rural areas than urban areas [6,10]. Even in the global context, rural areas across the world experience poorer health outcomes, lesser healthcare access and lower health literacy [11,12,13]. Therefore, thorough review of the existing research in Australia will strengthen the evidence of geographical difference and demographic characteristics between the rural and urban population and assist in prioritising strategies to improve mental health outcomes of Australians.

The annual expenditure of Australian healthcare system on mental disorders is $10.4 billion [14], making it the fourth most expensive disorder in Australia. Depression and anxiety are the most common mental health conditions affecting 1 and 2 million Australians annually [15], respectively. Moreover, depression and anxiety are the second and the third highest causes of burden of disease, respectively among the age group of 15–25 contributing to 7.6% and 7%, respectively to the total burden of diseases in that age group in Australia [5]. This is the scenario with many high-income countries [16]. Globally, high income countries spend more than 1% of their healthcare budget on mental health (in contrast to low- and middle-income countries that spend about 1% only on mental health), and yet have high prevalence of mental health conditions [16].

Although depression and anxiety affect people of all ages [17,18,19,20,21,22], the risk of developing mental disorders is more likely in youth (age group 10–24 years) [20,21,23,24,25]. Besides their high prevalence, depression and anxiety are a major concern because of their early onset during adolescence, and progression into adulthood affecting quality of lives and productivity [26]. The high prevalence and early onset among adolescents could largely be explained by the changing environment, biological changes such as puberty and sexuality, and environmental changes such as peer victimisation, smoking, alcohol, substance use [27,28,29,30,31]. When young people are affected by mental ill-health such as depression and anxiety, it not only affects them individually, but the broader society [32]. This ripple effect is because depression and anxiety are associated with poor school performance, school dropouts, substance abuse followed by unemployment and poverty, and suicide [1,33,34].

To address mental ill-health, there have been many intervention programs aimed at young people in Australia to help them recover from depression and anxiety [35,36,37,38]. While many yielded promising results, prevalence of depression and anxiety remains high amongst young people population. This gap in availability of intervention programs and yet a high prevalence in depression and anxiety can be explained by the lack of self-awareness of the symptoms resulting in many young people not being diagnosed [7,39]. It is important to understand the issue might not be the efficacy of these programs but help-seeking attitude [40,41,42,43]. One of the primary factors contributing to the lack of self-awareness or help-seeking attitude is the difference in levels of social determinants [6], mental health literacy and cultural acceptance of depression and anxiety as health conditions between rural and urban Australians [39,44,45,46]. However, there is conflicting research on the difference in prevalence of depression and anxiety between rural and urban Australians, especially in the age group of 15–24 years [44,46,47].

This manuscript focussed on Australian literature to systematically review the evidence base and strengthen knowledge. The focus on Australia is due to the evident geographical difference between the rural and urban population health outcomes [27,48] and demographic characteristics. An overview of Australian health status reveals that Australians living in rural and remote areas have shorter lifespans, poorer healthcare access and use, higher levels of disease and injury compared to people living in urban areas. However, unlike general health outcomes, there is a gap in research in understanding if geographical location; urban-rural dwelling has any effect on mental health conditions outcomes. If geographical location is a factor of mental health prevalence, then this could play an important role in policy implications towards customised programs and effective resource allocations that would suit the specific population living in the location (rural-urban locations in this case).

It is essential to investigate the differences in prevalence between rural and urban areas, not just from a societal standpoint but from an economic point of view as well. Taxpayers pay 33 times more towards treating psychological distress in urban areas than in rural areas [49]. Not all psychologists in Australia are covered by the Medicare and hence must charge a gap fee (Australian Psychological Society) [49]. This indicates a skewed spending of taxpayers’ dollars, and studies or reports that merely report the mental health prevalence rates would underrepresent the rural population. Studies examining the prevalence of mental health conditions, depression, and anxiety specifically, in Australian rural and urban areas are predominantly focused on specific cohorts, or individual contexts. No systematic review and meta-analysis have combined all qualifying studies to give a broader understanding of the Australian scenario in geographical variation of depression and anxiety among adolescents. Therefore, the primary aim of this paper is to assess the prevalence of depression and anxiety among rural and urban Australians in the age group of 10–24 years of age through systematic review and meta-analysis, to help identify geographic and demographic issues associated with mental health literacy and mental healthcare access in Australia.

## 2. Methods

The systematic review was conducted according to the Preferred Reporting Items for Systematic Reviews and Meta-Analysis (PRISMA) guidelines [50]. The review protocol was registered in PROSPERO International Prospective Register of Systematic Reviews in December 2020 (ID: CRD42020223600). Two authors (SK and VO) conducted the study selection, quality assessment and data extraction to avoid any potential biases. The other authors (TN, IM, JBG) contributed through expert knowledge, review and critical feedback on the systematic review and meta-analysis.

### 2.1. Search Strategy

All the articles for the systematic review were extracted through electronic search. The articles were searched from MEDLINE, PsychINFO, CINAHL, Web of Science and Scopus databases. A thorough citation search for additional articles was also undertaken. The final search for the articles was performed on 15 October 2022. The keywords used to filter the relevant studies were (“Anxiety” OR “Depression”) AND (“Rural” OR “Remote” OR “Isolated” OR “Regional” OR “Non-urban” OR Urban OR “City”) AND (“Youth” OR “Young Adult*” OR “Adolescent*” OR “Teenage*” OR “Young People”) AND (“Australia*” OR “Victoria” OR “New South Wales” OR “Queensland” OR “Western Australia” OR “Northern Territory” OR “South Australia” OR “Tasmania” OR “Australian Capital Territory”). Different combinations of keywords were searched to avoid missing any article.

### 2.2. Inclusion and Exclusion Criteria

Studies that had the following criteria were included: (1) Reported on either depression or anxiety or either depression or anxiety (we did not screen for specific types of depression or anxiety); (2) measured the prevalence of disorder either through percentages, or at least provide enough information to calculate prevalence of depression or anxiety (number of young people in sample with depression or anxiety, and total number of young people in the sample); (3) had a specified place of residence- either rural (remote and regional too) or urban and/or comparing both; (4) included a cohort between the ages of 10 and 24 years or provided separate breakdown of prevalence by age group or at least included an age range covering a portion of the required age cohort for this study; (5) used established depression or anxiety assessment tool through self-report screening or interviews; (6) published during the time period of January 2000–October 2022; (7) that were peer-reviewed journal articles. We excluded qualitative studies, study protocols and review manuscripts.

### 2.3. Study Selection

The author SK developed the key terms and screened titles and abstracts using the selection criteria. The author VO then independently screened all the articles meeting the selection criteria. There was a 95% agreement between the reviewers. For papers that did not have clear abstracts, the authors reviewed the full text. Full texts of short-listed studies were reviewed by SK and VO and inclusions were discussed till a conclusion was achieved. Authors TN, IM and JBG provided advice on this study.

### 2.4. Quality Assessment

All the included studies were assessed for risk of bias using the tool from “Methodological Guidance for Systematic Reviews of Observational Epidemiological Studies Reporting Prevalence and Incidence Data”, specifically designed for systematic reviews on prevalence studies [51]. Risk bias through nine domains were assessed using the tool: (1) Was the sample frame appropriate to address the target population? (2) Were study participants sampled in an appropriate way? (3) Was the sample size adequate? (4) Were the study subjects and the setting described in detail? (5) Was the data analysis conducted with sufficient coverage of the identified sample? (6) Were valid methods used for the identification of the condition? (7) Was the condition measured in a standard, reliable way for all participants? (8) Was there appropriate statistical analysis? (9) Was the response rate adequate, and if not, was the low response rate managed appropriately? The rating on each item is a four-point score of “yes” (low risk of bias), “no” (risk of bias), “unclear” and “not applicable”. The authors assessed each of the domains with an individual score of “yes”, “no”, “unclear” or “not applicable”, and then assigned a total score to each paper based on the assessment on the individual domains.

### 2.5. Data Extraction

From the included studies, the data was extracted using the following standard form: (1) study details such as title, date, author, study setting and study design (including the sample technique); (2) participant characteristics: age, gender; (3) outcome: screening tool, response rate, reported prevalence of depression and/or anxiety, any rural-urban areas differences and conclusion. For the meta-analysis, the data was formatted in the following form: title, authors, year of publication, sample age group, disorder measured, place of residence, sample size (denominator), reported number of those with depression or anxiety (numerator), and prevalence (calculated if not provided). Data was extracted by SK and reviewed by VO.

### 2.6. Meta-Analysis

Random effects model using DerSimonian and Laird method was fitted to the data [52]. The inverse of the Freeman–Tukey double arcsine transformation was used to stabilize the variance of each study [53]. Forest plot with the individual study prevalence estimates and the 95% confidence interval were presented for each region (rural vs. urban) and disorder (anxiety vs. depression vs. ‘depression or anxiety’) in subgroup analyses and the overall pooled prevalence estimates. The Z-statistic was used to test the subgroup and overall effect. Heterogeneity across studies was calculated using the *I*^2^ statistic. An *I*^2^ score of 25% represents low heterogeneity, 75% represents high heterogeneity and 50% represents medium heterogeneity [54]. Further, visual inspection of the Begg’s funnel plot, Egger’s test for small study effects and the non-parametric trim-and-fill analysis were used to assess for publication bias [55,56,57]. For all the analyses, *p*-Value ≤ 0.05 was considered as statistically significant. In meta-analyses, the *p* value provides information on the statistical significance of an effect, while the practical significance is explained by the effect size, which indicates if the effect is large enough to provide a meaningful insight to the population [58]. All analyses were conducted using Stata version 16.1 (StataCorp, College Station, USA) using the metaprop command [52].

## 3. Results

The literature search yielded 559 articles published between 1 January 2000 and 15 October 2022. Additionally, citation search of qualified papers resulted in extraction of 259 studies making it a total of 818 studies. After removing 142 duplicates, the final number of studies was 676 studies. After screening the title and abstracts, 33 studies qualified for full—text assessment. None of the citation search papers qualified for the full-text assessment. The entire PRISMA diagram of the study search can be found in Figure 1. Both the researchers independently reviewed the full—texts of the 33 articles (Figure 1). From the full—text screening, 10 articles [28,45,59,60,61,62,63,64,65,66] (references [28,45,59,60,61,62,63,64,65,66] are cited in the Appendix A) qualified for quantitative meta—analysis. Of the 10 studies, 2 studies [64,66] calculated the prevalence of either depression or anxiety, or prevalence of depression and anxiety together, 5 studies focused only on depression [28,45,59,61,65], 2 studies focused only on anxiety [60,63] and 1 study focused on depression and anxiety separately [62]. With respect to region, 3 articles [59,62,64] reported prevalence of depression and anxiety in rural area, 1 study [61] in urban area, and 6 articles [28,45,60,63,65,66] reported prevalence of depression and anxiety in both urban and rural areas. This resulted in 17 estimates for calculating pooled prevalence in the meta-analysis.

### 3.1. Study Characteristics

The meta-analysis identified 7650 participants from a total of 10 published studies all based in Australia. Of these 10 studies, 3 studies estimated depression prevalence in youth in urban and rural areas (6 separate estimates) [28,45,65], 2 studies estimated anxiety prevalence in urban and rural areas (4 separate estimates) [62,63], 1 study estimated depression prevalence in rural areas [59], 1 study estimated depression prevalence in urban areas [61], 1 study estimated either depression or anxiety or both in urban and rural areas (2 separate estimates) [66] and 1 study estimated either depression or anxiety or both in rural areas [64], and 1 study estimated anxiety and depression prevalence in rural areas (2 separate estimates) [60]. This provides a total of 17 estimates for meta-analysis purposes.

Although a few of studies included ages outside of the range that the study was aiming at—like 5–12 years [63], 4–10 years [60] and 15–85 years [45], the reviewers included the studies given that they covered a fraction of the target age-group population (10–24 years) (Appendix A).

All the included studies were of cross-sectional design. Five studies [59,61,63,65,66] used non-random sampling study design, wherein the participants were those admitted to mental health facilities for being at risk of suicide, which makes the sampling technique purposeful, where the questionnaire was presented to the mothers to answer and return, or the schools provided the questionnaire to the students to self-administer. The rest of the studies used random sampling techniques. Four studies were conducted in South Australia, of which three studies focused only on rural South Australia. Another four studies focused on New South Wales, both urban and rural/regional areas of Sydney such as Illawarra, Orange, Macquarie and Western Sydney, while the remaining studies sampled the rural and urban areas all over population of Australia.

All the studies used valid mental health screening tools, some were self-report and others were diagnostic tools designed to identify the presence of symptoms of depression and anxiety. Two studies used the sub-set of those attending mental health and/or suicide prevention facilities. All the studies adopted survey-based study design and none of the studies used estimates of prevalence through clinically diagnosed interviews. The Kutcher Adolescent Depression Scale (KADS), Spence Children’s Anxiety Scale – Parent Version (SCAS), the Strengths and Difficulties Questionnaire – Parent Version (SDQ), Rosenberg Self-Esteem Scale (RSES), Revised Child Anxiety and Depression Scale–Child (RCADS-C-25) and Revised Child Anxiety and Depression Scale–Parent (RCADS-P-25) Child Report of Parent Behaviour Inventory (CRPBI), Social Capital Questionnaire, Revised Child Anxiety & Depression Scales (RCADS), Beck Hopelessness Scale (BHS), the Depression Anxiety and Stress Scale— short version (DASS21–DEP), Interpersonal Needs Questionnaire (INQ), Adolescent Screening Questionnaire, the Primary Care Evaluation of Mental Disorders (PRIME-MD) which allows individual identification of DSM-III-R, Assessment of Quality of Life (AQoL) were the assessment tools used across all the studies.

The studies that sampled only participants with diagnosed depression and/or anxiety were chosen from Psychological Assistance Service for urban populations, and Centre for Rural and Remote Mental Health, for rural populations. For the remoteness, all the studies comparing urban and rural areas or simply focusing on rural areas used Accessibility and Remoteness Index of Australia (ARIA) categories to categorise remote/rural and regional areas from urban areas. One study used Socio-Economic Indexes for Areas (SEIFA) from ABS to categorise remote/rural and regional from urban areas. Quality assessment to identify any risk of bias is presented in Table 1.

### 3.2. Studying Forest Plots

Forest plots give us information about the effect sizes, heterogeneity, proportion and weight that is given to each study that forms the sample of the meta-analysis. The vertical line represents the null effect. Each study has a box plot with a horizontal line across. The midpoint of the box plot represents the point estimate, which explains the effect size of the study. When the study sample is larger, the box is bigger, and the horizontal line is shorter than otherwise. The horizontal line representing the confidence interval (CI 95%), which specifies that the boundaries within which the true value lies with 95% confidence. If the horizontal line crosses the line of no effect, it means the study is not illustrating a statistically significant result. The diamond represents the pooled estimate of the independent studies.

### 3.3. Prevalence of Depression and Anxiety

From the meta-analysis, we derived forest plots, which were deemed appropriate to report the results [67,68]. The forest plot (Figure 2) provides first the context for the meta-analysis, with individual studies’ data, namely, the effect size and its associated 95% confidence interval (95%CI), the relative weight assigned to each effect for computing the combined effect (i.e., summary statistic). The ES and 95%CI are also presented schematically. The effect sizes, in the forest plot, of depression ranged from 6.1% to 61.1%. For anxiety, effect sizes ranged from 19.6% to 39.1%, while for depression and anxiety together, effect sizes ranging from 25% to 40%. From the meta-analysis, the pooled prevalence of depression was 21.3% (95% CI, 14.9–28.5%) (Figure 2). The pooled estimate of anxiety was 29.9% (95% CI, 21.6–39.0%). For both depression and anxiety together, the pooled prevalence estimate was 27.2% (95% CI, 20.3–34.6%). These pooled effect sizes estimates are represented numerically (with their associated 95% CI) and schematically (in green diamonds) as well on the forest plot (Figure 2). From the *I*^2^ score, a measure of the magnitude of heterogeneity among studies (see Figure 2), it can be deduced that there exists high between-study heterogeneity *(I*^2^
*=* 94.7%, *p* < 0.001).

Subgroup analysis: The heterogeneity between groups was significant (*p* < 0.001), while the overall heterogeneity between all the studies (depression, anxiety and depression or anxiety) was high (*I*^2^ = 96.3%, *p* < 0.001). Since each study used a different scale to measure depression, anxiety and/or both, it is not possible to check the subgroup prevalence of similar assessment tools.

### 3.4. Rural-Urban Areas Prevalence of Depression and Anxiety

The pooled prevalence of depression and anxiety for urban areas was 26.1% (95% CI, 17.3–35.9%) (Figure 3). The effect sizes ranged from 6.1% to 44.7%. The pooled prevalence for depression and anxiety in rural areas was 24.9% (95% CI, 17.5–33.3%), while individual estimates ranged from 8.2% to 61.1%. High heterogeneity between studies is evident for both rural subgroup (*I*^2^ = 94.8%, *p* < 0.001), urban subgroup (*I*^2^ = 97.0%, *p* < 0.001) and all studies together (*I*^2^ = 95.8%, *p* < 0.001). However, heterogeneity between the rural subgroup and urban subgroup is not statistically significant *(p =* 0.85).

The test for risk of bias: The Begg’s funnel plots appear slightly asymmetric (Appendix A), but the Egger’s test did not provide any evidence of publication bias (*p* = 0.20) as seen in Figure 4. Additionally, after imputing one study to correct for asymmetry using the trim-and-fill analysis, the pooled prevalence (25.3%, 95% CI: 18.7–31.2%) following the imputation did not differ from our original finding (Appendix A).

### 3.5. Sensitivity Analysis

The first sensitivity analysis was conducted by removing the three studies that reported very large ranges of age groups: 5–12 years [63] and 15–85 years [45] and 4–18 years [60]. Results show that removal of these studies did not change the pooled prevalence estimates (Appendix A).

The next sensitivity analysis was conducted by removing studies that did not have random sampling; studies that used sample of people already at high risk of suicide [61] or psychosis [66]. With this sensitivity analysis, the pooled prevalence of depression was 17.8% (95% CI, 13.1–23.0%) (Appendix A). From the *I*^2^ score, it can be deduced that there exists high between-study heterogeneity (*I*^2^ = 89.4%, *p* < 0.001). The pooled estimate of anxiety did not change from the previous meta-analysis. Furthermore, the overall pooled prevalence estimates of anxiety and depression is 23.2% (95% CI, 17.9–28.9%).

Subgroup analysis: The heterogeneity between groups was significant (*p* < 0.05), while the overall heterogeneity between all the studies was high (*I*^2^ = 96.0%, *p* < 0.001). (Appendix A)

### 3.6. Rural-Urban Areas Prevalence of Depression and Anxiety

The pooled prevalence of depression and anxiety for urban areas was 21.2% (95% CI, 13.0–30.7%). The pooled prevalence for depression and anxiety in rural areas was 24.6% (95% CI, 17.0–33.2%). High heterogeneity between studies is evident for both rural subgroup (*I*^2^ = 95.4%, *p* < 0.001), urban subgroup (*I*^2^ = 97.0%, *p* < 0.001) and all studies together (*I*^2^ = 96.0%, *p* < 0.001). However, heterogeneity between the rural subgroup and urban subgroup is not statistically significant *(p = 0.56)*. (Appendix A)

This study contributes evidence on mental health outcomes in countries with geographical heterogeneity in physical health outcomes such as Australia. The study aimed to primarily estimate the prevalence rates of youth mental ill-health in Australia (overall and by rural/urban grouping). As the meta-analysis aims to synthesis and pull out the summary or combined effect, here the prevalence rate, and its measure of precision (captured in the 95% confidence interval), the findings reveal new estimates of the prevalence rates of youth mental ill-health overall and in rural/urban settings. The subgroups analysis also intends to explore any potential rural/urban differences in pooled prevalence rates of depression and anxiety.

The absence of significant difference in rural/urban youth prevalence is, in itself, a positive finding. Moreover, the pooled estimates (rural: 24.9% vs. urban: 26.1%), although not statistically significant, should not be qualified as “no difference between the sub-groups” or “no new finding”.

## 4. Discussion

This systematic review and meta-analysis of 10 studies with 17 estimates involving 7650 participants aged between 10 and 24 years across Australia, examining differences between urban and rural populations shows higher but nonsignificant prevalence of depression and/or anxiety in urban areas (26.1%) than in rural areas (24.9%) of Australia. From the meta-analysis, the pooled prevalence of depression was 21.3% and anxiety was 29.9%. National estimates by the ABS National Study of mental health and wellbeing [48], show that in the age group of 16–25 years, almost one third (31.5%) of Australians experience anxiety disorders in a 12-month period and 13.6% experienced depressive disorders. These ABS prevalence estimates for anxiety are almost identical to those found in our study, whereas our estimate for depression is higher than the national estimate. This could be explained by our inclusion of non-random sample studies, however, with sensitivity analyses conducted to check for any bias, note that the prevalence of depression and anxiety do not substantially alter in subgroup analyses. Given the lack of national estimates comparing the prevalence of depression and anxiety between urban and rural regions in Australia amongst the 10–24-year-old group, this meta-analysis provides the first evidence of prevalence and direct comparison for depression and anxiety in rural and urban areas of Australia in this age group.

This study finds a slight difference in urban-rural prevalence among young people in Australia, which is consistent with the findings from previous studies that compared urban-rural mental health outcomes among young people [45,62,69] However, these studies also found that the use of anti-depressants is more common among young rural people than their urban counterparts, indicating a difference in the management of mental health conditions. Studies estimating prevalence of depression and anxiety among young people in UK [70] USA [71] and China [72] have concluded that place of dwelling (rural-urban) does not affect prevalence of depression, consistent with the findings of our meta-analysis. However, there are some Australian studies that estimate that rural populations have higher rates of mental health conditions than urban population [73,74].

On the other hand, a few studies in Canada showed that young urban people had a higher prevalence rate of depression than their rural counterparts [75,76,77]. Several studies have concluded that community belongingness in rural areas and lack of competition that urban youth often face are the reasons for lower prevalence rates in rural areas [76,77]. However, another study [75] also found that rural young people are less likely to access mental health services.

There are various other factors that affect the mental health outcomes such as socioeconomic status, cultural differences, employment opportunities, exposure to mental health literacy, healthcare access especially mental health services, education levels and biological factors [27,29,30,31,39,44].

This study has provided scope to potentially explain the overall high prevalence of mental ill-health despite the high mental healthcare expenditure. One aspect for further research particularly applicable to the youth population is the influence of social media. With the rapid onset of social media, there has been inconclusive research findings on the influence of social media on mental well-being, with many researchers concluding that it depends on how it is being used [78]. Research also shows that illustrating empathy and care instead of judgement in social media platforms has resulted in positive mental well-being of individuals [79]. One of the ways this can be mediated is through awareness spread by social media influencers, who serve as pacesetters in the community, thereby establishing a personal and psychological relationship with their consumers [80]. The influence of social media on adolescent mental health is a highly debated topic and hence must be explored further, particularly in the context of the rural-urban divide.

In this review and meta-analysis, not all the studies provided the gender prevalence separately, there were almost equal number of male and female participants from the nine studies. From these ten studies, one study reports a dramatic increase in prevalence of depression among females (23%) compared to males (11.8%) from the age of 15 years to 18 years [59]. In this study, they found no significant difference between ARIA+ scores and females who screen depressed versus females who did not. However, another study revealed that rural boys reported higher depression compared to rural girls and urban boys and girls [65]. The rest of the studies reported no significant difference in prevalence of depression and/or anxiety between males and females. Existing literature suggests that female gender is a risk factor for depression and anxiety [81,82,83,84], and that women seek help [59,65] and take anti-depressants [50] more than males. Furthermore, studies found that young urban [50,81] and rural females [59,85,86] experience higher rates of depression, anxiety and suicide [85] than their male counterparts. However, some studies focusing on rural areas suggest that males, especially young men, experience more depression and other mental health conditions than their female counterparts [6,65,87].

Two of ten studies used for the meta-analysis studied the population who are at high risk of suicide. This indicates that every participant in that cohort has an established mental health condition, depression in most cases. Besides, most of the studies used in the review and meta-analysis used non-random sampling techniques, which could lead to selection bias and over-estimation of the prevalence of the disorders. This could have resulted in prevalence estimates being higher than the national estimates.

Nevertheless, our estimates show a considerably high prevalence rates of depression and/or anxiety in the young people population of Australia with the urban population having a slightly higher prevalence than rural areas. There is not enough literature measuring the rural-urban population difference in prevalence of depression and anxiety in the young people population. This also resulted in selecting only 10 studies out of the 676 studies. The number of studies for depression were much higher than for anxiety and both disorders together, and the studies for rural were higher than the studies for urban areas. The limited literature could be an issue of low awareness as discussed above. Therefore, more inquiry into depression and/or anxiety in rural versus urban Australia will advance the knowledge of the geographical disparities in depression and/or anxiety among young people as well as inform prevention and management practices.

Most studies used surveys provided to schools or parents, they were not administered by the researchers in person. This led to some schools choosing the students sample, parents answering questions for their children and many teenagers self-administering the survey. This is necessary because directly contacting the cohort population (aged between 10 and 24) would have not been permitted without the intervention of schools or parents. One study [88] provided literature evidence that the assessment administering technique does not change the result, i.e., it is not a concern to the research findings if the screening was done through face-face delivery, remote or online. However, the authors [88] suggest that with non-random sampling techniques, non-face-to-face delivery of assessment techniques would only increase the selection bias, and that future research must use face-to-face methods wherever possible to mitigate the risk of bias.

## 5. Strengths and Limitations

The biggest strength of this review is the meticulous planning and search strategy following the PRISMA guidelines closely resulting in thorough work. The search terms were precise, and all relevant synonyms were used which ensured that potential articles were not missed. Another strength of this review is that only peer-reviewed and published articles were searched for and large databases such as Scopus were reviewed.

All studies used different assessment tools, study designs, specific cohorts which may influenced the variability of the prevalence estimates. Additionally, the non-random sampling techniques used for more than half of the studies could result in selection bias. Only 2 studies compared prevalence of both depression and anxiety between rural and urban population. This made it impossible to explore the interaction between rural vs. urban and depression vs. anxiety.

## 6. Conclusions

The results of this study show that a quarter of Australian young adults aged between 10 and 24 years have depression and/or anxiety. About 26.1% in the urban areas and 24.9% in the rural areas reported prevalence of depression and/or anxiety. While there is ample literature evidence that young people are at the highest risk of developing mental health conditions, and rural Australia experiences lower health outcomes, research continues to be inconclusive in terms of comparing the prevalence. This is largely hindered by the mental health illiteracy and help-seeking attitude coupled with poor healthcare access and social determinants in the rural areas of Australia. Despite many successful and effective programs to diagnose and treat depression and anxiety, and a big proportion of healthcare budget allocated to treat mental health conditions, Australia continues to have high prevalence of mental health conditions, depression and anxiety precisely. The response from policy makers and service providers must be more inclusive to promote better healthcare access and educate people on mental health. This paper assessed prevalence of depression and anxiety among rural and urban Australians aged 10-24 years using a systematic review and meta-analysis to identify potential geographic patterns in mental ill-health, with implications for mental health literacy and access to healthcare in Australia. This study also provides insights into the applicability of using this methodology more widely across countries with similar geographical and demographic composition in the context of public health.

## Figures and Tables

**Figure 1 ijerph-20-00800-f001:**
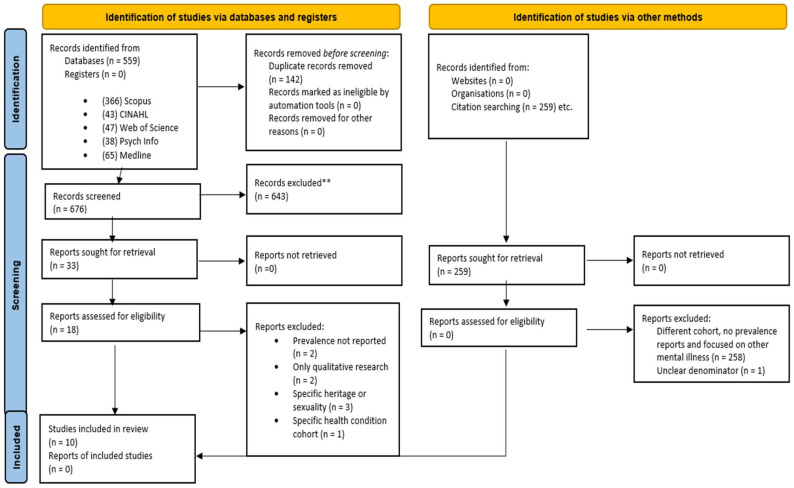
PRISMA flowchart (2020).

**Figure 2 ijerph-20-00800-f002:**
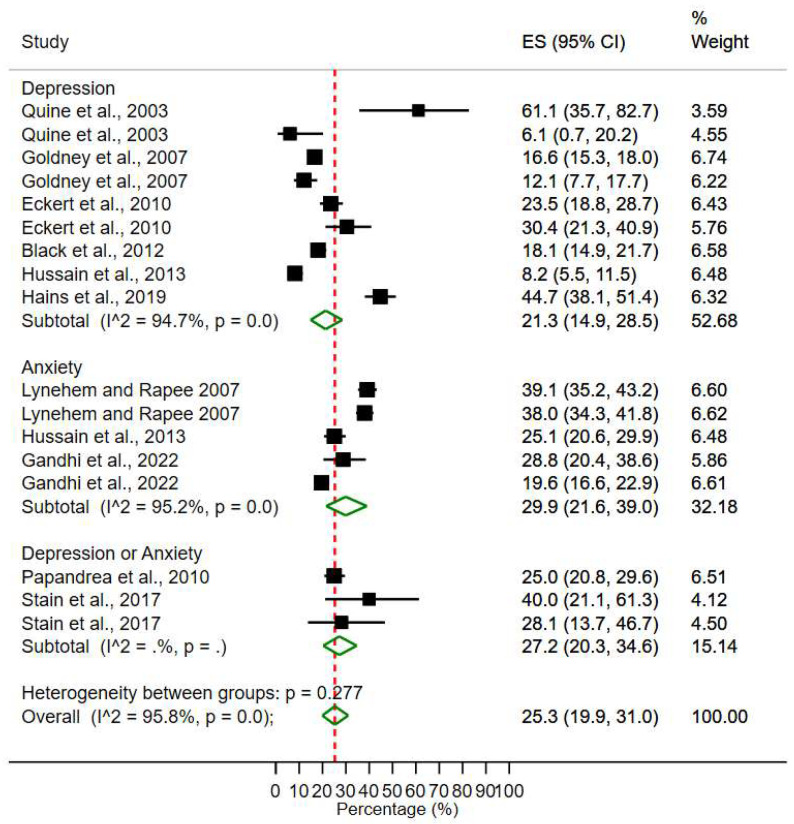
Meta-analysis and pooled estimate of depression, anxiety and both disorders [14,28,45,59,60,61,62,63,65,66].

**Figure 3 ijerph-20-00800-f003:**
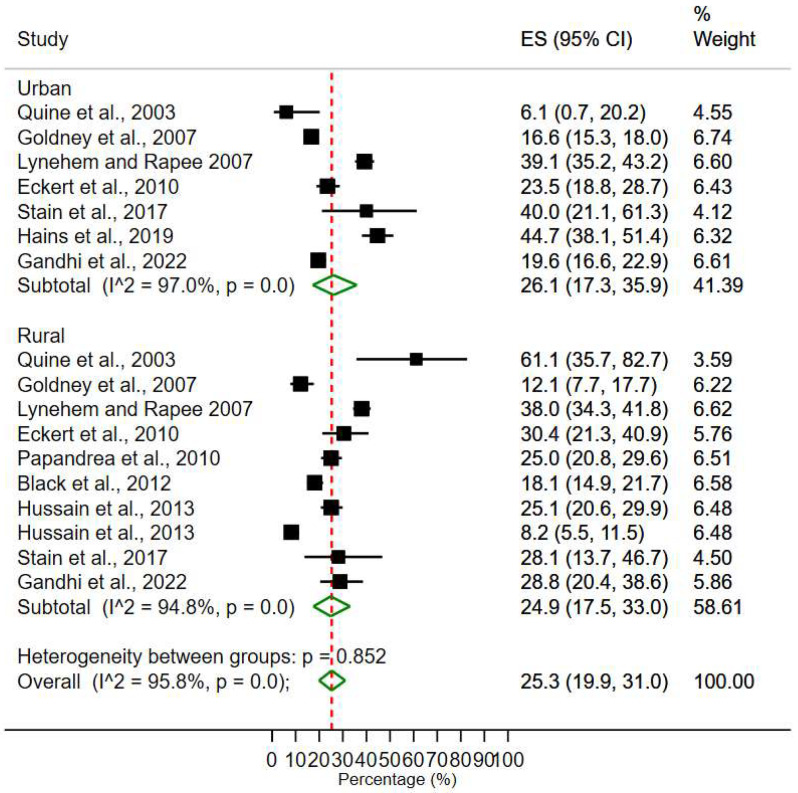
Meta-analysis and pooled estimate of urban area and rural areas [28,45,59,60,61,62,63,64,65,66].

**Figure 4 ijerph-20-00800-f004:**
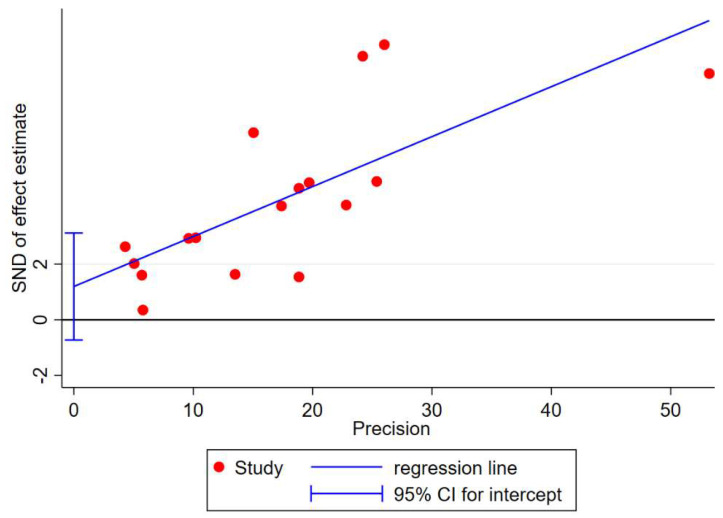
Egger’s test of studies evaluating potential publication bias.

**Table 1 ijerph-20-00800-t001:** Quality assessment results of the 10 studies included in the meta-analysis.

Study	Q1	Q2	Q3	Q4	Q5	Q6	Q7	Q8	Q9	Total
Lynehem and Rapee, 2007 [63]	Y	Y	Y	Y	Y	Y	Y	Y	Y	9
Hussain et al., 2013 [62]	Y	N	Y	Y	Y	Y	Y	Y	Y	8
Papandrea et al., 2010 [64]	Y	Y	Y	Y	Y	Y	Y	Y	Y	9
Stain et al., 2017 [66]	Y	Y	N	Y	Y	Y	Y	Y	N	7
Quine et al., 2003 [65]	Y	Y	Y	Y	U	Y	Y	U	Y	7
Black et al., 2012 [59]	Y	Y	Y	Y	N	Y	Y	Y	N	7
Goldney et al., 2007 [45]	Y	Y	Y	Y	Y	Y	Y	Y	Y	7
Eckert et al., 2010 [28]	Y	Y	N	Y	Y	Y	Y	Y	Y	8
Hains et al., 2019 [61]	N	N	Y	Y	Y	Y	Y	Y	Y	7
Gandhi et al., 2022 [60]	Y	Y	N	Y	Y	Y	Y	Y	N	7
Total Y	9	8	7	10	8	10	10	9	7	
Total N	1	2	3	0	1	0	0	0	3	
Total U	0	0	0	0	1	0	0	1	0	
**Y%**	**90%**	**80%**	**70%**	**100%**	**80%**	**100%**	**100%**	**90%**	**70%**	

## Data Availability

The data presented in this study are available in the references below: 10,17,22,24,27,31,36,46,48,54 and in the Appendix A.

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
