# Peer review of "Prevalence Rates of Depression and Anxiety among Young Rural and Urban Australians: A Systematic Review and Meta-Analysis"

_ijerph, 2023, doi:10.3390/ijerph20010800_

Round 1

Reviewer 1 Report (Previous Reviewer 1)

Dear authors,

thank you for revising your manuscript "Prevalence rates of depression and anxiety among young rural and urban Australians: A systematic review and meta-analysis.".

However, the authors should try to incorporate their answers into the manuscript for the below-mentioned, rather than putting them exclusively to the reviewer:

Introduction:

- Sources should be brought up-to-date. How can an introduction with outdated resources give the motivation for a whole research manuscript of today?
(the authors added three sources, one from the year 2000, which is also outdated)

- It remains unclear & seems to be subjective to draw the found research gap from the given information.

Findings: 

- The manuscript lacks scientific research. The boxplots give a descriptive overview on the assessed studies. However, the data from the boxplots is just described. It is not used, nor is there any other information given in the manuscript.

I further wish the authors all the best.

Best regards

Author Response

Document addressing reviewer’s comments

Reviewer#1

Comment 1: Introduction: - Sources should be brought up-to-date. How can an introduction with outdated resources give the motivation for a whole research manuscript of today? (the authors added three sources, one from the year 2000, which is also outdated).

- It remains unclear & seems to be subjective to draw the found research gap from the given information.

Response: We thank the reviewer for highlighting this aspect. The sources in the introduction and other sections (except for some technical references) have now been updated to reflect newer sources corresponding to the research. Please see references 7, 13, 14, 19, 23, 38, 50, 81 on the manuscript.

The purpose of this research was to study existing literature through a systematic review and perform a meta-analysis to combine the results of these studies to make the existing evidence stronger. The existing literature can be studied in the background context of three themes that explain the gap and purpose of this research: 1) the availability of mental health services in Australia such as intervention programs [1,2,3, 11], mental health literacy [4] and professional help [5] and the high healthcare budget allocated for mental health treatment and services [6], 2) the evident geographical variation in health outcomes and healthcare access between urban and rural Australia [7,8], yet uncertain differences in prevalence of mental health conditions, 3) the high prevalence and early onset of mental health disorders among young population. These are the status quo aspects of health across Australia.

While it is established that rural areas have poorer healthcare outcomes and healthcare access compared to their urban counterparts [7,8], despite high healthcare budget [6], there is a gap in research especially in studying geographical variation of common mental health conditions such as depression and anxiety are major public health concerns globally. We know that Australia has a high prevalence of mental health conditions [9, 10], and the young people are at the highest risk (and adolescence is the age-of-onset of mental health conditions) [10, 11,12], but unlike general health outcomes, there is a gap in research in understanding if geographical location has any effect on mental health conditions outcomes. If geographical location is a factor of mental health prevalence, then this could play an important role in policy implications towards customised programs and effective resource allocations that would suit the specific location (rural-urban in this case). With our study, we found that there is very small difference between urban and rural prevalence of mental health conditions. The area of mental health among young people in a geographical context needs further research and meta-analysis is the starting point of this work. The lack of difference is a finding which means there is need for more research.

We have revised the manuscript to justify this inclusion (Page 2 lines 41-52; page 12, lines 24-34; page 13, lines 1-2).

Comment 2: The manuscript lacks scientific research. The boxplots give a descriptive overview on the assessed studies. However, the data from the boxplots is just described. It is not used, nor is there any other information given in the manuscript.

Response: We thank the reviewer for this comment. The purpose of this research was to systematically review existing literature in the field of mental health in the young people population of Australia and check if geographical location of dwelling had an impact on the prevalence rates of common mental health disorders, depression and anxiety in this young people population. This has been performed through meta-analysis which combined the estimates from the existing literature that focussed on certain rural and urban populations of Australia. Since there is no systematic review conducted in this specific research, our study attempted to combine the estimates for strong evidence. Existing literature is focussed on specific cohorts of specific regions of Australia, which is why we have used these estimates to make a conclusion for Australia as a whole.

After receiving this comment, we have made and corrected in the manuscript that the meta-analysis used forest plot for analysis and not a boxplot (corrected in the manuscript). While chose forest plots because they are the best way to identify a common statistic from a group of studies used for the study. The black box in the forest plot represents the point estimate of the study. The forest plot depicted the required statistics (effect size, CI, pooled estimates and heterogeneity) which we have written about in the results section.

References:

  1. Johnston L.; Dear B, F.; Gandy, M.; Fogliati, V,J.; Kayrouz, R.; Sheehan, J.; Rapee, R, M.; Titov, N. Exploring the efficacy and acceptability of Internet-delivered cognitive behavioural therapy for young adults with anxiety and depression: An open trial. Aust N Z J Psychiatry 2014, 48, 9, 819–827. DOI: https://doi.org/10.1177/0004867414527524
  2. Long, J,R. The acceptability of using MOODGYM to treat depression in adolescents: a pilot study. Graduate Student Theses, Dissertations, & Professional Papers 2016, 10898. Available online: https://scholarworks.umt.edu/cgi/viewcontent.cgi?article=11947&context=etd
  3. Newcombe, P,A.; Dunn, T,L.; Casey, L,M.; Sheffield, J,K.; Petsky, H.; Anderson-James, S.; Chang, A,B. Breathe Easier Online: evaluation of a randomized controlled pilot trial of an Internet-based intervention to improve well-being in children and adolescents with a chronic respiratory condition. J Med Internet Res 2012, 14, 23. Doi: https://doi.org/10.2196/jmir.1997
  4. Hernan, A.; Philpot, B.; Edmonds, A.; Reddy, P. Healthy minds for country youth: help-seeking for depression among rural adolescents. Aust J Rural Health 2010, 18, 118–24. Doi: https://doi.org/10.1111/j.1440-1584.2010.01136.x
  5. Blau, A.; Byrd, J.; Piper, G. Far from Care: How your postcode can influence whether you need help — and if you’ll get it. Available online: https://www.abc.net.au/news/2020-12-08/covid-mental-health-system-medicare-inequality/12512378 (accessed on 12 June 2021).
  6. Australian Institute of Health and Welfare. Disease expenditure in Australia 2018- 19. Available online: https://www.aihw.gov.au/reports/health-welfare-expenditure/disease-expenditure-australia/contents/summary (accessed on 30 September 2021).
  7. Australian Institute of Health and Welfare. Rural and remote health. Available online: https://www.aihw.gov.au/reports/rural-remote-australians/rural-and-remote-health (accessed on 10 July 2022).
  8. Smith, J. A., Canuto, K., Canuto, K., Campbell, N., Schmitt, D., Bonson, J., Smith, L., Connolly, P., Bonevski, B., Rissel, C., Aitken, R., Dennis, C., Williams, C., Dyall, D., & Stephens, D. Advancing health promotion in rural and remote Australia: Strategies for change. Health Promot J Austr 2022 33,1, 3–6. Doi: https://search.informit.org/doi/10.3316/informit.289662503616295
  9. Black Dog Institute. Facts & figures about mental health. Available on: https://www.blackdoginstitute.org.au/wp-content/uploads/2020/04/1-facts_figures.pdf?sfvrsn=8 (accessed on 10 August 2020).
  10. Australian Bureau of Statistics. National Survey of Mental Health and Wellbeing, 2020-21. Available online: https://www.abs.gov.au/statistics/health/mental-health/national-study-mental-health-and-wellbeing/latest-release (accessed on 03 July 2022).
  11. Gautam, M.; Tripathi, A.; Deshmukh, D.; Gaur, M. Cognitive Behavioral Therapy for Depression. Ind J Psychiatry 2020, 62, 2, S223-S229. doi: 10.4103/psychiatry.IndianJPsychiatry_772_19.
  12. Martin, J.; Hadwin, J ,A. The roles of sex and gender in child and adolescent mental health. JCPP Adv 2022 2, e12059. DOI: https://doi.org/10.1002/jcv2.12059

Reviewer 2 Report (Previous Reviewer 2)

The manuscript has been significantly improved and can be considered for  publication in ijerph.

Round 2

Reviewer 1 Report (Previous Reviewer 1)

Dear authors,

thank you for allowing me to review your revised manuscript. 

Introduction:

- While Australia's characteristics are explained sufficiently, I would propose to also make a link to the global characteristics to show the significance of the study.

Formal:

- Table 1: A more than 1-side-long table should be placed in the appendix, even if it is the core of the manuscript.

- Please check grammar + formatting (e.g. not in block) in the added text.

I wish the authors all the best

Author Response

Document addressing reviewer’s comments

Reviewer#1_Round 2

Comment 1: Introduction: - While Australia’s characteristics are explained sufficiently, I would propose to also make a link to the global characteristics to show the significance of the study.

Response: We thank the reviewer for this comment. The introduction has now been updated to include the global scenario of this issue (page 1 lines 29-34; page 2, lines 13-15 and 25-28)

Comment 2: Formal- A more than 1-side-long table should be placed in the appendix, event if it is the core of the manuscript

  • Please check grammar +formatting (e.g., not in block) in the added text.

Response: We thank the reviewer for this comment. The table is now moved to the appendix. Please find Table T1 on the supplementary file. The paper has now been checked and corrected for grammar and formatting errors.

This manuscript is a resubmission of an earlier submission. The following is a list of the peer review reports and author responses from that submission.

Round 1

Reviewer 1 Report

Dear authors,

thank you for giving me the chance to read through your review paper "Prevalence rates of depression and anxiety among young rural and urban Australians: A systematic review and meta-analysis.".

With the post-COVID era in sight, I want to point out some points that shall help to further strengthen the paper:

Topic/Segment:

- Why should anybody outside Australia be interested in the study? The study focuses on Australia without a given scientific reason. Further, almost all literature, also in the discussion section seems to focus on Australia.

Abstract:

- Should be unstructured.

Introduction:

- Sources are outdated or even very much outdated. The limitation to Australian literature seems to hinder the authors to establish a level field to work on. 

- What is the motivation of the study? What is the gap to close?

Results:

- The meta-analysis results show some boxplots without any value to the reader. They present descriptive data that the authors do not work with in later stages. 

Findings: 

- The findings do not reveal anything new. There is no significant difference, even almost no difference between any segment. The authors do neither answer any review question nor do they seem to have the motivation to contribute to the scientific body of knowledge.

Discussion:

- In fact, the authors show that there is no new finding as the available literature base in Australia is very limited. 

Formal:

- Please check Enligsh typos & grammar, especially towards the end of the manuscript. 

- References are not according to journal standard. 

- Please consider whether big tables should not be put into an appendix (table 1). A short summary table could be sufficient in the text. 

- Different fonds used in the manuscript. 

- The author's information section should be acc. to journal style.

I wish the authors all the best.

Best regards

Reviewer 2 Report

The authors explored the prevalence rates differences of depression and anxiety among young rural and urban Australians by meta-analysis and explained the possible reasons for the differences.

Q1: The value of the study is not convincing enough. Why do we need to understand differences of prevalence rates depression and anxiety among young rural and urban areas through meta-analysis , since it has been discussed in many previous studies. What is the specificity of this study ? The current study does not seem to show the value of meta-analysis.

Q2: Authors points that “there is conflicting research on the difference in prevalence of depression and anxiety between rural and urban Australians, especially in the age group of 15-24 years (Eckert et al., 2010; Marshall and Dustan, 2013; Zhuang et al., 2017).”  But what is the reason for the authors' final selection of the study sample in 10-24 age group? Is it reasonable to choose a sample that spans different life stages (adolescents, emerging adults, adults)?

Q3 The meta-analysis in this study involved only 10 studies, and if depression and anxiety were explored separately, the sample size would be even smaller; does this sample size accurately explain the differences? Are there relevant studies that would support the small sample discussion in a meta-analysis?

Q4: The description of the studies in the manuscript is somewhat confusing.

For example: The literature search yielded 559 articles published between 1st January 2000 to 15th October 2022. Additionally, citation search of qualified papers resulted in extraction of 259 studies. After removing 142 duplicates, the final number of studies was 676 studies.  So 259 to 677 studies?

Table 1. Base characteristics of the 9 studies included in the meta-analysis.

There are 7650 participants from a total of 10 studies all based in Australia.

So 10 or 9 studies?

Q5: The discussion should be based on the study findings. This study did not find significant differences of prevalence rates depression and anxiety among young rural and urban Australian. The current discussion is too general and does not correlate well with the study's conclusions

Reviewer 3 Report

The study provides meaningful implications for academics and practitioners. However, the authors are recommended to present the importance of social-media messages in affecting anxiety and depression. For example, messages shared by social-media influencers (SMIs) are influential in evoking individuals' positive emotions (Cheung et al., 2022). Social media community facilitates sharing amongst individuals and thereby improve their wellbeing (Kross et al., 2021; Yang et al., 2020)

The authors are recommended to cite these studies and discuss the importance of social media messages, to improve the quality of the paper. 

Cheung, M. L., Leung, W. K., Aw, E. C. X., & Koay, K. Y. (2022). “I follow what you post!”: The role of social media influencers’ content characteristics in consumers' online brand-related activities (COBRAs). Journal of Retailing and Consumer Services66, 102940.

Kross, E., Verduyn, P., Sheppes, G., Costello, C. K., Jonides, J., & Ybarra, O. (2021). Social media and well-being: Pitfalls, progress, and next steps. Trends in Cognitive Sciences25(1), 55-66.

Yang, Y., Liu, K., Li, S., & Shu, M. (2020). Social media activities, emotion regulation strategies, and their interactions on people’s mental health in COVID-19 pandemic. International Journal of Environmental Research and Public Health17(23), 8931.